# Apolipoprotein A-I (ApoA-I), Immunity, Inflammation and Cancer

**DOI:** 10.3390/cancers11081097

**Published:** 2019-08-01

**Authors:** Konstantina Georgila, Dimitra Vyrla, Elias Drakos

**Affiliations:** 1Department of Biology, Medical School, University of Crete, Heraklion, Voutes, 71110 Crete, Greece; 2Department of Pathology, Medical School, University of Crete, Heraklion, Voutes, 71110 Crete, Greece

**Keywords:** apolipoprotein A-I, HDL, cancer, immunity, inflammation, review

## Abstract

Apolipoprotein A-I (ApoA-I), the major protein component of high-density lipoproteins (HDL) is a multifunctional protein, involved in cholesterol traffic and inflammatory and immune response regulation. Many studies revealing alterations of ApoA-I during the development and progression of various types of cancer suggest that serum ApoA-I levels may represent a useful biomarker contributing to better estimation of cancer risk, early cancer diagnosis, follow up, and prognosis stratification of cancer patients. In addition, recent in vitro and animal studies disclose a more direct, tumor suppressive role of ApoA-I in cancer pathogenesis, which involves anti-inflammatory and immune-modulatory mechanisms. Herein, we review recent epidemiologic, clinicopathologic, and mechanistic studies investigating the role of ApoA-I in cancer biology, which suggest that enhancing the tumor suppressive activity of ApoA-I may contribute to better cancer prevention and treatment.

## 1. Introduction

Apolipoprotein A-I (ApoA-I), the major protein component of high density lipoprotein (HDL), widely known for regulating cholesterol trafficking and for protecting against cardiovascular disease (CVD), may also modulate inflammatory and immune responses [1]. Recent studies suggest that organismal metabolic changes that include shifts in the levels and the quality of ApoA-I, may facilitate cancer initiation and progression [2,3]. Herein, we present and review the findings of various epidemiologic, clinicopathologic, and mechanistic studies implicating ApoA-I in cancer, with emphasis on its connection with inflammatory and immune-modulating effects.

The *ApoA1* gene is regarded to have the same evolutionary origin with the genes of apolipoproteins A-II, A-IV, C-I, C-III, and E, by virtue of duplication and diversification of a basic genetic motif encoding an 11/22 amino acid sequence with a characteristic α-amphipathic helix signature [4,5,6,7]. Homologous ApoA-I-encoding genes have been described in mammals, birds, and teleost fish [8]. 

The regulation of human *ApoA1* gene expression is complex and is controlled at multiple levels. The transcription of human *ApoA1* largely depends on two hormone response elements (HREs) proximal to the transcription start site that bind members of the hormone nuclear receptor superfamily. Among them, peroxisome proliferator-activated receptor-γ (PPARγ) appears to have a prominent role in *ApoA1* transactivation by interacting with HREs as heterodimer with RXRα. Other transcription factors implicated in the regulation of *ApoA1* promoter include the hepatocyte nuclear factor 4 (HNF4), Liver Receptor Homologue 1 (LRH1) and the ApoA-I Regulatory Protein 1 (ARP1/NR2F2) which activate and repress the *ApoA1* promoter, respectively [9]. HNF4 operates together with Sp1 in the communication of *ApoA1* promoter with enhancer sequences that facilitate the recruitment of the basal transcriptional machinery. 

ApoA-I expression is also controlled by a long noncoding RNA, *ApoA1-AS*, which is transcribed in the apolipoprotein gene cluster on chromosome 11q23.3 and modulates suppressive epigenetic marks leading to *ApoA1* transcriptional repression [10]. Interestingly, the liver, small intestine, and colon where ApoA-I is predominantly detected, show approximately 100-fold higher expression levels of *ApoA1* mRNA compared to *ApoA1*-AS, whereas the *ApoA1*/*ApoA1*-AS ratios are less than one in most other tissues [10]. Post-transcriptional mechanisms may also contribute to the regulation of ApoA-I expression in certain conditions. Thus, an enrichment of polysomal fractions with *ApoA1* mRNAs explains the increase in ApoA-I synthesis observed in high fat-fed mice in the absence of an effect on transcription [11].

Following translation and intracellular removal of a N-terminal signal peptide, ApoA-I is secreted as a lipid-poor/free mature protein of 243 amino acids and a molecular weight of approximately 28kDa [6]. Its structure contains ten consecutive helical regions, critical for the biophysical properties of the protein to spontaneously solubilize lipids in aqueous environment [6]. Based, exactly, on the properties of these amphipathic helical motifs, various peptides, without sharing any sequence homology, have been synthesized, known as ApoA-I mimetics, because of their ability to simulate ApoA-I functionality [12,13,14]. In physiological conditions, the bulk of ApoA-I constitutes approximately 70% of the protein component of HDL, which are microemulsions composed of a nonpolar lipid core, a surface polar lipid monolayer and up to 95 different proteins [15,16].

HDL are heterogeneous and dynamic structures exchanging lipids with cells and other lipoproteins, classified to different subcategories with pre-β1 HDL corresponding to lipid-poor ApoA-I [17,18]. ApoA-I is essential for the assembly of HDL. ApoA-I stabilizes the ATP-binding cassette transporter 1 (ABCA1), a member of the ABC superfamily, at the cell membrane of hepatocytes and enterocytes, enabling it to mediate the efflux of cellular phospholipids and free cholesterol to nascent discoid HDL particles harboring two to four molecules of ApoA-I, leading to the biogenesis of HDL particles. A similar lipid efflux by ABCA1 in cells of peripheral tissues initiates the reverse cholesterol transport (RCT) [17,19] (Figure 1). Also, ApoA-I activates lecithin cholesterol acyl transferase (LCAT), leading to the maturation of HDL particles [20]. Interaction of lipidated ApoA-I in discoid or more mature HDL particles with another transporter of the ABC family, ATP-binding cassette subfamily G member 1 (ABCG1), contributes further to the RCT [21]. HDL particles undergo additional remodeling through interaction with the cholesteryl ester transfer protein (CETP) [22]. Finally, binding of HDL particles to the scavenger receptor class B type 1 (SR-BI), transfers cholesterol down a cholesterol gradient [23,24]. As a result, cholesterol mobilized at peripheral tissues can enter the liver and is catabolized and excreted to the bile [24,25,26]. ApoA-I itself is mainly catabolized in the liver [27,28] (Figure 1). 

Besides promoting RCT, ApoA-I inhibits apoptosis and pro-oxidative and proinflammatory processes in endothelial cells, induces vasodilation, inhibits the activation of platelets, and contributes to innate immunity. Some of these functions are relevant to inflammatory and malignant processes and are discussed below.

## 2. ApoA-I, Immunity, and Inflammation

Throughout its evolutionary course, ApoA-I/HDL contributes to the humoral part of innate immunity [29]. It has antiviral activity associated with prevention of viral penetration, facilitation of complement-mediating bacterial killing, and protection against trypanosome brucei, a protozoal parasite [30,31,32]. ApoA-I protects from sepsis by binding to and neutralizing lipopolysaccharide (LPS) and lipoteichoic acid (LTA), components of the Gram-negative and Gram-positive bacterial cell wall, respectively [33,34]. Clearance of LPS through binding of HDL-LPS to SR-BI results in lower activation of the Toll-like receptor 4 (TLR4), the corresponding pathogen-associated molecular pattern (PAMP) recognition receptor, and in decreased production of tumor necrosis factor (TNF), interleukin 1β (IL-1β) and interleukin 6 (IL-6) by the proinflammatory cells that mediate sepsis pathology [29,35,36]. In line with these experimental findings, reduced serum ApoA-I levels in sepsis patients are associated with poor prognosis [37,38]. Also, ApoA-I was found to increase the levels of pentraxin 3 (PTX3), an acute phase protein, which recognizes PAMPs in viruses, bacteria, and fungi [39,40].

Inflammatory cytokines such as TNF and IL-1β repress the production of ApoA-I from hepatocytes and increase the expression of serum amyloid A (SAA), which becomes the major protein component of HDL in this context [41,42,43]. Consequently, lipid-poor ApoA-I is rapidly catabolized in the liver and the kidney. These findings could be meaningful if ApoA-I, in addition to its proimmune features, had anti-inflammatory potential. In this way, removal of ApoA-I could intensify the inflammatory response, resulting in a more robust effect. On the other hand, decreased levels of ApoA-I could contribute to destructive chronic inflammation characterizing many autoinflammatory and autoimmune diseases. Indeed, a plethora of studies have shown that ApoA-I exhibits anti-inflammatory features by various mechanisms [1]. In the context of the humoral arm of innate immunity, it has been shown that ApoA-I inhibit the formation of the terminal attack complex of the complement, C5b-9, by interfering with C9 polymerization and incorporation into the membrane and contributes to complement clearance [44]. Also, ApoA-I-mediated increase of PTX3 levels could contribute to a better healing, given that PTX3 can promote efficient tissue repair [45]. 

In a seminal study, it was shown that mice deficient for the receptors Abca1 and Abcg1 display marked leukocytosis and a transplantable myeloproliferative disease, which can be suppressed by transgenic overexpression of ApoA-I [46]. These findings suggest an inhibitory role of ApoA-I in cellular components of the immune system which has been postulated to relate to the lipid-modulating function of ApoA-I. One potential mechanism involves modulation of cholesterol-enriched lipid raft microdomains that function as docking sites for several receptors, coreceptors, and costimulatory molecules in neutrophils, monocytes/macrophages, dendritic cells (DC), and B and T lymphocytes [47,48,49]. ApoA-I, via ABCA1, reduces the abundance of lipid rafts and lowers the levels of CD11b expression leading to downregulation of neutrophil activation, migration, and adhesion [50]. A similar mechanism has been proposed for downregulation of TLR signaling in macrophages and major histocompatibility (MHC) class II molecule expression in antigen presenting cells with consequent attenuation of adaptive immune responses [51]. Inhibition of dendritic cell maturation and differentiation by ApoA-I is associated with elevated secretion of prostaglandin E2 (PGE2) and IL-10 and downregulation of IL-12 and IFN-γ [52]. Similarly, inhibition of dendritic cell maturation and downregulation of Th1 and Th17 cell reactivity by ApoA-I/HDL leads to attenuation of arthritis in an antigen-induced murine arthritis model [53]. 

In contrast, a recent study provided evidence that the ability of ApoA-I/HDL to suppress the TLR-mediated secretion of proinflammatory cytokines IL-6 and TNF in monocytes was dependent on transcriptional events mediated by the induction of activating transcription factor 3 (ATF3) and independent of TLR signaling and cholesterol modulation in lipid rafts, implying “outside-in” signaling events induced by ApoA-I/HDL that remain obscure [54]. Another anti-inflammatory mechanism was recently proposed. It was observed that ApoA-I/HDL decreased the expression of inflammasome components, including NLR family pyrin domain containing 3 (NLRP3) and IL-1β, as well as caspase 1 activation in human macrophages [55]. In addition, by using a murine model of atherosclerosis, it was shown that myeloid Abca1/g1 deficiency enhanced caspase-1 activation in monocytes, macrophages, and neutrophils, resulting in enhanced atherogenesis that was suppressed by Nlrp3 or Caspase-1/11 deficiency [56,57]. Also, the link between ApoA-I/HDL and inflammasome activation in dendritic cells and has been recently reported in a systemic lupus erythematosus-like murine model [58]. These findings suggested that accumulation of cholesterol in macrophages, or dendritic cells acting as a “danger signal”, could activate the inflammasome leading to chronic inflammation, something that can be opposed by ApoA-I/HDL.

Additional effects on specific cellular compartments of the immune system by ApoA-I have been discovered. Thus, administration of ApoA-I suppressed inflammation in autoimmune-prone mice lacking both LDL-receptor and ApoA-I, an effect that was associated with expansion of regulatory T cells (Treg) and a decrease of effector/effector memory T cells [59]. Another study showed that ApoA-I and ABCA1 play a pivotal role in the extracellular release of isopentenyl pyrophosphate and the consequent activation of Vγ9Vδ2 T cells, a specialized type of lymphocytes that recognize phosphor-antigens in a TCR-dependent but MCH-independent manner [60].

Deregulated immunity against microorganisms and pathogenic chronic inflammation can be viewed as different aspects of the same process. A recent study showed that mice deficient in ApoA-I exhibit exaggerated colitis, while administration of an ApoA-I mimetic peptide attenuated gut inflammation, which was associated with decreased secretion of IL-6 by epithelial enterocytes in response to LPS, abundant in the gut lumen [61]. In agreement with these findings, another study reported that the intensified chemically-induced colitis observed in the setting of selective deletion of transcription factor EB (Tfeb) in the murine intestinal epithelium was associated with reduced ApoA-I expression [62]. It has been suggested that part of the anti-inflammatory properties of ApoA-I/HDL may be due to its contribution to innate immunity mechanisms including its ability to neutralize bacterial products [61]. Given that chronic inflammatory conditions may predispose to various types of malignancy, the anti-inflammatory effects of ApoA-I may impinge on cancer-related processes as discussed below in Section 6.

It must be added that the anti-inflammatory properties of ApoA-I depend not only on the levels of the protein but also on its functionality [63,64,65]. Many epigenetic alterations of ApoA-I, including oxidative modifications, observed in chronic inflammation may erase its anti-inflammatory features or even transform it to a proinflammatory agent [66].

## 3. A Potential Protective Role of ApoA-I against Cancer: Evidence by Association

Accumulating evidence suggests that regulation of the ApoA-I/HDL axis is derailed in cancer. Our mining of transcriptome microarray data registered in the Oncomine database (https://www.oncomine.org) and of recently published RNAseq data [67] uncovers reduced *ApoA1* mRNA levels in hepatocellular carcinoma (HCC) compared to normal liver tissue, the main source of ApoA-I. The transcriptional repression of *ApoA1* in HCC remains mechanistically unexplored but it is in line with the reported reduction in protein levels of ApoA-I in both cancerous liver tissue [68] and in the serum of HCC patients [69,70]. HDL itself is also reduced in HCC [71]. Collectively, the reduction in *ApoA1* transcription, intracellular and secreted ApoA-I, and circulating HDL levels in HCC hint to a putative tumor suppressor role of this pathway. Indeed, numerous studies have discovered associations between the levels of serum ApoA-I/HDL and various parameters of the natural history of many types of cancer (summarily presented in Table 1).

The Alpha-Tocopherol, Beta-Carotene (ATBC) cancer prevention study showed inverse association between HDL-associated cholesterol (HDL-c) levels and the risk for the development of lung, liver, and hematologic malignancies [72]. The Women’s Health Study investigating the cancer risk in female health workers, found that lower levels of HDL were associated with higher risk for the development of lung and colorectal cancer [73]. The Malmo Diet and Cancer Study revealed an inverse association between the risk for the development of colorectal, lung, and breast cancer and the levels of HDL-c and ApoA-I [74]. The correlation between lower levels of HDL/ApoA-I and higher risk for colorectal cancer was also reported in a Korean cross-sectional study, while premalignant lesions of colorectal cancer (colon adenomas) were shown to be associated with lower HDL levels in a cohort of patients examined by colonoscopy [75,76]. Similar associations have been reported for prostate cancer by a Swedish cohort study and for Hodgkin and non-Hodgkin lymphoma by the Cancer Research Network lymphoma study [77,78]. The latter found that the more pronounced drop in HDL levels was observed 3–4 years prior to lymphoma diagnosis [77]. 

In line with these risk association studies, reduced serum levels of HDL/ApoA-I have been reported in cancer patients at first diagnosis, indicating that HDL/ApoA-I may be a potential biomarker for early cancer detection. A study analyzing serum lipid profiles of patients diagnosed with any type of solid tumor and healthy controls showed decreased HDL/ApoA-I levels, specifically, in the cancer group [140]. Similarly, relatively decreased levels of HDL/ApoA-I have been reported in many cancers of the gastrointestinal tract including adenocarcinomas of the stomach, the colon, the pancreas, and hepatocellular carcinoma (HCC) [70,76,83,92,93,94,98,104,105]. A serum proteomic analysis of patients with chronic liver disease associated with hepatitis C virus (HCV) infection showed that the development of HCC was associated with lower levels of ApoA-I [69]. Also, relatively reduced serum HDL/ApoA-I levels have been found in patients with lung and breast adenocarcinoma, early stage ovarian and cervical cancer, and acute lymphoblastic leukemia [84,106,117,118,119,120,126,136].

The levels of HDL/ApoA-I have also been associated with the progression of the neoplastic disease and the response to therapy. Reduced serum ApoA-I levels correlate with the progression of lung, liver, breast, kidney, endometrial, and cervical cancer, associated with the appearance of metastases [99,107,123,124,125,127,137,141]. A postoperative serum proteomics analysis of high-risk breast cancer patients also showed that low expression of ApoA-I was associated with metastatic relapse [107,134]. Other studies reported that ApoA-I serum levels were significantly decreased in HCC patients with recurrent disease, as compared to patients in remission, and patients with acute lymphoblastic leukemia who achieved remission after receiving chemotherapy displayed significant increases in ApoA-I levels [100,137].

An overall association of HDL/ApoA-I levels with the prognosis of cancer patients treated with surgery, chemotherapy, radiotherapy, or immunotherapy has been concluded in a recent meta-analysis [142]. Indeed, an association with prognosis has been reported in patients with nasopharyngeal carcinoma, non-small cell lung carcinoma, invasive breast ductal adenocarcinoma, esophageal squamous cell carcinoma, colorectal adenocarcinoma, HCC, renal cell carcinoma, and transitional cell carcinoma of the bladder [80,81,82,85,86,88,89,95,101,102,108,109,135,143]. Likewise, ApoA-I has been proposed as a putative prognostic biomarker in neuroblastoma patients, since ApoA-I serum levels were found significantly lower in patients with high risk tumors [138]. Interestingly, post-treatment ApoA-I levels also seem to confer prognostic significance. A retrospective study of colorectal cancer patients treated with surgery and adjuvant chemotherapy showed that relatively increased levels of HDL-c and ApoA-I, one month after the completion of chemotherapy, were associated with better prognosis [144]. ApoA-I levels have been especially evaluated in response to chemosensitivity. Higher serum ApoA-I were found to be associated to better response to chemotherapy in patients with colorectal cancer, while higher ApoA-I levels secreted in the interstitial fluid of breast tumors were associated with more chemosensitive tumors [145,146]. In another study, ApoA-I levels were found to predict response to IMA901, the first therapeutic vaccine used in a randomized phase 2 trial for the treatment of patients with advanced renal cell carcinoma. Specifically, high levels of ApoA-I were associated with better overall survival [147].

Possible associations of ApoA-I genetic variations with cancer parameters have also been noted. A positive association was found between the ApoA-I (−75) A allele and breast cancer risk, and between the ApoA-I (+83) T allele and the development of lymph node metastasis [110]. Another study showed that breast cancer patients carrying an ApoA-I-rs670 A allele showed a less favorable phenotype at presentation, with absence of hormone receptor expression and lymph node metastases in comparison to G/G carriers. Moreover, rs670 A/A carrying patients had more frequent recurrences and inferior survival in comparison to patients with no A alleles [111].

Although the majority of studies have shown an inverse association of ApoA-I levels with the development and progression of various cancers, positive correlations have been reported. For example, ApoA-I levels are upregulated in the serum of patients with early stage gastric adenocarcinoma, recurrent head and neck squamous cell carcinoma and retinoblastoma and in the urine of patients with transitional cell carcinoma of the bladder, while a nested case-control study reported that HDL-c/ApoA-I levels were positively associated with the risk for the development of breast cancer [79,90,112,113,130,131,139]. Also, pro-ApoA-I levels were found upregulated in the serum of lung cancer patients with brain metastases and overexpressed at transcriptional level in metastases of colon adenocarcinoma to the liver suggesting that, in these particular situations, ApoA-I levels could be used as a biomarker for the extension of the disease to the brain and the liver, respectively [87,96,148]. It is unclear whether the positive correlation between ApoA-I levels and cancer parameters reported in a minority of studies are specific reflecting, in these particular situations, tumor promoting processes, associated with increased cholesterol uptake of malignant cells through the HDL/SR-BI pathway, as proposed by some studies [149,150,151].

Even though the bulk of data suggest that increased levels of ApoA-I/HDL could be protective against cancer development and progression, the reported ApoA-I/HDL alterations could be a consequence and not a cause of the carcinogenesis process. In such a case, ApoA-I could still be a useful biomarker for early cancer detection, or for better management stratification and follow up of cancer patients. However, a causative role would imply that interventions aiming at increasing the levels and functionality of ApoA-I could contribute to better cancer prevention and therapy.

## 4. ApoA-I Exhibits Tumor Suppressive Activity: Evidence from In Vitro Studies

A number of in vitro studies suggest that ApoA-I affects the proliferative, survival, and migratory behavior of various carcinoma cells, largely through cell-autonomous mechanisms (summarily presented in Table 2).

HCC cells treated with recombinant ApoA-I undergo G0/1 cell cycle arrest and apoptosis associated with downregulation of mitogen-activated protein kinases 1 and 3 (MAPK1, MAPK3), known for their antiapoptotic function, and upregulation of proapoptotic genes including caspase 5 (casp5), tumor necrosis factor receptor superfamily 10B (TNFRSF10B), and apoptosis protease activating factor 1 (APAF-1) [100]. ApoA-I also induced downregulation of vascular growth factor (VEGF) and matrix metalloproteinases 2 and 9 (MMP2, MMP9) genes, suggesting that ApoA-I may decrease the angiogenetic potential and the ability of HCC cells to remodel extracellular matrix, inhibiting in this way their metastatic potential [100].

A recent study showed that human colon adenocarcinoma (CA) cells stably transfected with ABCA1, exhibit increased proliferative, invasive and migratory behavior, which could be inhibited by simultaneous, transgenic overexpression of ApoA-I, or by exogenous treatment with human recombinant ApoA-I [157]. This inhibition was associated with downregulation of cyclooxygenase 2 (COX-2), a known promoter of colon adenocarcinoma involved in proinflammatory processes. In the same study, apabetalone, a small molecule BET-inhibitor, used in experimental therapeutics of atherosclerosis and known to induce production of ApoA-I, reduced the ABCA1-driven proliferative and invasive behavior of CA cells [157]. Another study showed that treatment of CA cells with the ApoA-I-mimetic peptide L-4F induced G0/1 cell cycle arrest, associated with decreased expression levels of cyclins D1 and A and decreased cell viability [158]. Also, it reduced the survival of CA cells stimulated by lysophosphatidic acid (LPA), a potent bioactive phospholipid, known to decrease its free concentration in the cell culture media [158]. 

In ovarian carcinoma (OC) cells, treatment with ApoA-I-mimetic peptides D-4F or L-4F was also found to impact proliferation, survival, and migratory behavior associated with reduced lipid peroxidation and hydrogen hyperoxide levels, and to decrease VEGF production and expression of the hypoxia induced factor-1α (HIF-1α) transcription factor [152,153,155,156]. Moreover, administration of ApoA-I or various ApoA-I mimetic peptides increases the sensitivity of human OC cells to cisplatin, a classical chemotherapeutic agent, associated with decreased activation of AKT [154]. Also, the ApoA-I mimetic D-4F was shown to reduce the proliferative response of human breast adenocarcinoma cells, stimulated by oxidized low-density lipoprotein (oxLDL) [159].

## 5. The Tumor Suppressive Function of ApoA-I: Evidence from Animal Studies

Accumulating evidence suggests that ApoA-I inhibits the growth of tumors and the metastatic progression of the disease in various animal cancer models (summarily presented in Table 3).

In a melanoma model, mice deficient for ApoA-I showed increased tumor burden and reduced survival. Conversely, transgenic overexpression of human ApoA-I or exogenous administration of ApoA-I protein reduced malignant burden, decreased metastases and increased mouse survival [161]. Melanomas in transgenic mice expressing high levels of ApoA-I showed decreased angiogenesis, reduced expression of MMP9, a matrix-degrading enzyme contributing to the invasive behavior of tumor cells, and reduced levels of survivin, an important antiapoptotic molecule. 

In line with the reported in vitro effects of ApoA-I mimetic peptides in OC, transgenic overexpression of human ApoA-I in a murine model of OC, or exogenous administration of D4-F, L5-F, and L4-F decreased tumor burden and increased survival [152]. The levels of LPA, VEGF, and HIF-1α in mice treated with ApoA-I mimetic peptides were found significantly reduced relative to control animals [152,155,156,163]. Another study showed that the inhibitory effects of the ApoA-I mimetic peptide D4-F was dependent on the upregulation of the antioxidant enzyme manganese superoxide dismutase (MnSOD), as silencing of the gene in the engrafted cells by a MnSOD-specific shRNA abolished the D4-F-tumor suppressing effects, suggesting that the antioxidant activity downstream of ApoA-I may be essential for its tumor suppressor properties in OC [153].

Patients with inflammatory bowel disease have increased risk for the development of CA [164]. In both humans with ulcerative colitis and mouse models of colitis-associated carcinogenesis, CA develops predominantly in the distal part of the large intestine. Intriguingly, ApoA-I^−/−^ and ApoA-I^+/−^ mice develop more numerous and larger tumors that display extension to the proximal part of the colon [61]. These differences were accompanied by a higher tumor cell proliferation rate in the ApoA-I^+/−^ group and by elevated expression levels of activated STAT3 [61], a transcription factor involved in inflammatory and tumor-promoting processes [165]. In another study, treatment with the ApoA-I mimetic peptide L-4F significantly reduced the size and number of polyps in Adenomatosis Polyposis Coli (APC)^−/+^ mice, a mouse model for human familial adenomatous polyposis [158]. Interestingly, the administration of ApoA-I mimetic peptides or the overexpression of ApoA-I not only reduced primary tumor burden but also metastasis of CA cells in the lung [161,162].

The tumor suppressive activity of ApoA-I has also been demonstrated in an orthotopically implanted mouse model of pancreatic adenocarcinoma and in a breast cancer mouse model, the latter being associated with reduction in plasma oxLDL [159,160]. Interestingly, dysfunctional, oxidized ApoA-I/HDL has been reported to promote breast cancer metastasis in mice [166]. Other animal studies have reported inverse association of serum ApoA-I levels with the progression of lung and gastric cancer in the mouse [167,168,169,170]. 

## 6. Anti-Inflammatory and Immune-Modulating Mechanisms Are Involved in the Tumor Suppressive Activity of ApoA-I

Collectively, the aforementioned in vitro and animal studies provide convincing evidence that ApoA-I affects many of the originally proposed hallmarks of cancer [171], including sustained proliferative signaling, resistance to cell death, angiogenesis, and activation of invasion and metastasis (Figure 2).

Accumulating findings suggest that ApoA-I also targets one of the more recently proposed hallmarks of cancer, that of tumor promoting inflammation. Chronic inflammation, caused by dysbiosis, plays a pivotal role in cancer promotion in the liver and colon [172,173], tissues known to produce ApoA-I, and inflammatory mechanisms emanating from TLR4 stimulation on cancer cells contribute to malignant growth in this context [172,174]. The concept that the anti-inflammatory properties of ApoA-I participate in the protection against colon cancer is highlighted by the fact that ApoA-I ameliorates colitis-promoted colon carcinogenesis in parallel with the attenuation of TLR4-mediated activation of key inflammatory regulators, including NF-_Κ_B, STAT3, and IL-6 [61]. This is further supported by studies demonstrating reduction of various oxidized lipids and enzymes involved in inflammation, such as COX-2, by ApoA-I in models of colon or ovarian cancer [156,157]. 

Given the role of ApoA-I/HDL in RCT (see Section 1), deregulation of this pathway may have systemic effects on lipid and cholesterol accumulation which, in turn, may impact on immune cell homeostasis and inflammatory reactions that are linked to malignancy [175,176]. Moreover, modulation of the integrity of cholesterol-enriched microdomains in the plasma membrane, which function as docking sites for several receptors, may alter the activation of signaling pathways in many cells of the immune system [47]. Additionally, ligation of HDL particles to specific ApoA-I receptors (ABCA1, ABCG1 etc.) promoting the RCT may result in broader “outside-in” signaling events which have been reported to enable macrophages to convey anti-inflammatory effects [177]. It is also possible that ApoA-I, internalized by the responsive cells may further modify signaling mechanisms. 

As the modulation of the inflammatory tumor microenvironment is exploited by cancer cells to buffer the attack of the immune system, the effects of ApoA-I on cancer inflammatory and immuno-editing processes seem interconnected [178,179]. Indeed, the ability of ApoA-I to inhibit melanoma growth is attenuated, although not abolished, in mice lacking the humoral and the cellular components of specific immunity [161]. Immuno-phenotyping of tumors developed in ApoA-I transgenic mice showed a reduction in myeloid derived suppressor cells (MDSCs), a heterogeneous immature myeloid cell population of granulocytic or monocytic origin capable of inhibiting the immune response, but an increase in tumor infiltrating cytotoxic T cells (TIL) and CD11b+ macrophages [161,180]. The latter is of particular interest as ApoA-I has been implicated in the conversion of tumor associated macrophages (TAM) from M2 to M1 phenotype that associates with enhanced antitumor properties [161]. Along these lines, administration of the ApoA-I mimetic peptide L4-F resulted in decreased recruitment of M2 macrophages to the tumors [160]. However, the exact mechanisms involved in the regulation of MDSC and M1/2 phenotype by ApoA-I remain elusive. 

One mechanism by which ApoA-I mimetic peptides impact on antitumor immunity entails an increase in the levels of specific oxidized lipids to activate Notch signaling in the intestine which, in turn, leads to higher numbers of patrolling monocytes in lamina propria. This treatment also reduces 25-hydroxycholesterol with concomitant decrease in osteopontin expression in enterocytes and lower numbers of MDSCs in lamina propria [162].

The notion that the tumor suppressive properties of ApoA-I are connected to the modulation of anticancer immunity is further supported by recent animal studies that investigated the effects of the ApoA-I receptors, Abcg1 and Abca1, on cancer growth in connection with parameters of anticancer immunity [181,182]. It was shown that myeloid specific deficiency of Abcg1 or Abca1 was associated with decreased tumor growth, increased polarization of TAMs towards M1 phenotype, and decreased numbers of specific MDSCs subsets in the tumors [181,182].

## 7. Tools for Therapeutic Targeting of ApoA-I

The tools for therapeutic targeting of ApoA-I originated from an effort to discover strategies exploiting the atheroprotective properties of HDL [183]. Some of these strategies aim to indirectly augment the ApoA-I and HDL-c levels by inhibiting endothelial lipase and CETP or to augment RCT by activating LCAT or the liver X receptors (LXRs), members of the nuclear receptor superfamily that orchestrate the activation of many genes promoting RCT and intestinal HDL production [183,184,185,186,187]. None of these approaches have been thoroughly tested in cancer studies. 

However, strategies aiming in directly augmenting ApoA-I, or mimicking ApoA-I functionality have already been used successfully in preclinical cancer studies (Table 2 and Table 3). The first of these approaches can be accomplished by intravenous administration of autologous delipidated HDL, purified native ApoA-I, or recombinant ApoA-I Milano protein, a mutated “hyperfunctional” ApoA-I variant discovered in a cohort of Italian patients, in complexes with phosphatidylcholine [188,189]. Although administration of such reconstituted HDL has shown antitumor activity in various preclinical models, it is a laborious and expensive strategy, difficult for a broad application in cancer patients.

Another approach utilizes ApoA-I mimetic peptides, synthesized on the basis of α-amphipathic helical repeating structure of ApoA-I, aiming to mimic the function of ApoA-I [190]. Many of them are 18 amino acids long and modified in various ways for augmenting stability and lipid-binding properties. Some of them including peptides composed of D-amino acids, being resistant to protease degradation, can be given orally, and have been expressed transgenically in tomatoes, in an effort to increase their practical utility [191]. Their action in small intestine tissues has been shown to be critical for their anti-atherogenic value in animal models [192]. These agents can produce HDL-like particles that promote cholesterol efflux and have shown antiatherogenic, antioxidant, anti-inflammatory, and antitumor activity in preclinical models [190,193]. Their function is not exactly equivalent to ApoA-I, since some are designed to better mimic one or another function of ApoA-I. For example, some of them have far superior ability, in comparison to ApoA-I, to neutralize pathogenic lipids such as LPA [190,192]. This may explain the discrepancy in the findings regarding alterations of LPA levels among various cancer studies using ApoA-I mimetics and ApoA-I [152,161] as well as differences in the antitumor activity [159].

Although ApoA-I mimetics have shown promising therapeutic potential in various preclinical models, recent clinical trials in the context of CVD have failed to demonstrate clear clinical benefit [194]. However, all clinical trials so far have been performed in the setting of acute coronary syndrome regarding patients with advanced disease in need for aggressive intervention. It is possible that future, carefully designed clinical trials, investigating a longer period of administration time in combination with established chemotherapeutic or immunotherapeutic agents, could be more informative for the therapeutic potential of ApoA-I mimetics in cancer.

## 8. Open Questions for Future Research

In vitro and in vivo experimental studies have shown that the tumor suppressive activity of ApoA-I targets cell-autonomous and cell-nonautonomous survival mechanisms (Figure 2). Which mechanism is pivotal for the antitumor activity of ApoA-I has not been fully elucidated. The prevailing view is that the anti-inflammatory action of ApoA-I is important for the enhanced antitumor immunity. Recent studies have shown that chronic inflammation is an essential mechanism contributing to the attenuation of innate and specific immunity against cancer [195]. Although there is evidence that ApoA-I may modify the anticancer immune response, detailed investigation of the effects of ApoA-I on immune checkpoints (for example programmed death ligand-1, PD-L1) in cancer cells or the cells of tumor microenvironment and on anticancer immunity has not been performed [161]. The effects of ApoA-I on the outcome of cancer immunotherapy also remains to be elucidated. 

Whilst the anti-inflammatory action of ApoA-I has attracted most attention, additional hallmarks of cancer may be influenced by ApoA-I. Findings showing that ApoA-I can affect the expression levels of transcription factors, such as HIF-1α, suggest that ApoA-I may have profound effects on metabolic pathways in cancer cells [156,196]. However, detailed alterations in energetic metabolism of cancer cells, including lipid metabolism, after treatment with ApoA-I or ApoA-I mimetic peptides have not been explored. The antioxidative function of ApoA-I has been demonstrated in various cancer models [153,156]. Although it is known that increased oxidation stress may contribute to DNA damage and increased mutational burden, the effects of ApoA-I on DNA damage response mechanisms have not been explored. ApoA-I itself is subject to oxidative damage and carbonylation which have been linked to apolipoprotein dysfunction and a pathogenic role in Alzheimer disease [197]. Whether these modifications may also have a role in immunity, inflammation and cancer remain obscure.

Most efforts to clarify the mechanism of the anti-inflammatory and antitumor activity of ApoA-I have focused on the interaction of ApoA-I with the receptors ABCA1, ABCG1, and SR-BI. For example, it has been shown that mice deficient in ABCG1 and ABCA1, when fed a “western”-type diet, display reduced growth of tumors derived from subcutaneously engrafted melanoma or bladder carcinoma cells, while other studies have attempted to associate ABCA1 with epithelial mesenchymal transition in breast cancer [181,182,198,199]. Although deficiency of ABCA1 or ABCG1 transporters does not always mirror the ApoA-I effect, at first glance, it seems counterintuitive that deficiency of ABCA1 and excess of ApoA-I, which affect RCT in opposite directions, both may contribute to tumor suppression. However, it is possible that some of the anti-inflammatory and antitumor activities of ApoA-I may be mediated by other receptors. For example, the beta-chain of ATP F1 synthase was discovered to represent a high affinity receptor of lipid poor ApoA-I at the cell membrane (known also as ecto-F1F0-ATPase; Figure 1) [25]. Binding of ApoA-I to this receptor was found to stimulate the hydrolysis of extracellular ATP to ADP and phosphate, implying that ApoA-I may affect signaling emanating from cell membrane P2 purinergic receptors, many of which have been shown to modify inflammatory and immune responses and tumor growth [26,200]. Although one of these receptors, P2Y13, was shown to mediate the signal from ecto-F1F0-ATPase to SR-BI for promoting HDL cell internalization, ecto-F1F0-ATPase-mediated effects of ApoA-I on P2 purinergic receptor signaling important for immune responses and cancer biology have not been investigated.

ApoA-I levels have been shown to be affected by chemotherapy. Examination of serum lipid profiles in breast cancer patients revealed a significant reduction of ApoA-I and HDL levels upon completion of chemotherapy [141,201]. ApoA-I protein levels were reduced by doxorubicin, while they remained unaffected by cyclophosphamide and paclitaxel treatment, in agreement with in vitro experimental findings [201]. Postchemotherapy infections are an important complication of cancer patients, and the ability of ApoA-I to neutralize bacterial products represents an important aspect of innate immunity [202]. However, the impact of ApoA-I on the incidence and outcome of postchemotherapy bacterial infections has not been investigated in epidemiologic or preclinical studies.

## 9. Conclusions

In conclusion, combined epidemiologic, clinicopathologic, and preclinical experimental research has shown that ApoA-I could represent not only a useful cancer biomarker, but a biochemical variable of the organism that could be modified for more effective cancer prevention and treatment. The exact mechanisms involved in the antitumor activity of ApoA-I and the evaluation of its antitumor therapeutic potential merits further investigations.

## Figures and Tables

**Figure 1 cancers-11-01097-f001:**
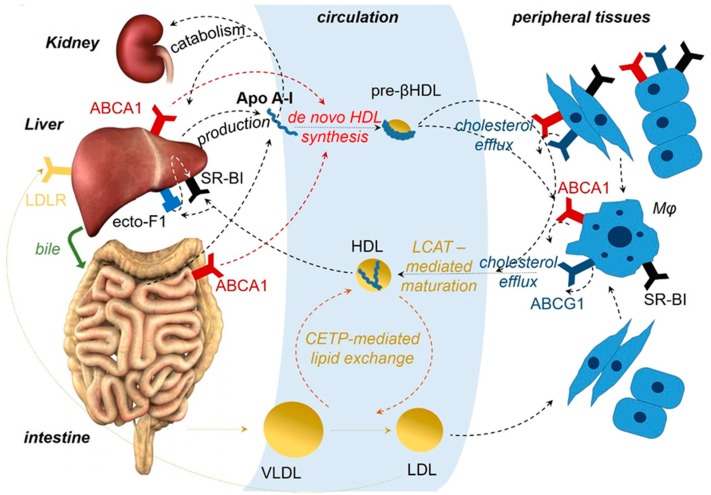
ApoA-I in relation to high-density lipoprotein (HDL) biogenesis and reverse cholesterol transport (RCT). About 75% of the ApoA-I protein is produced by hepatocytes and the remaining 25% by epithelial cells of the small intestine. It has been shown that some ApoA-I is also produced by the most proximal part of the mouse colon, in line with the reported ApoA-I expression in human fetal colon. ApoA-I is mainly catabolized in the liver. In addition, ApoA-I protein unassociated with lipids can be filtered in renal glomeruli, recognized by cubulin, a protein synthesized by distal renal tubular cells, internalized and degraded by renal epithelial cells. Binding of ApoA-I to ABCA1 at the cell membrane of hepatocytes and enterocytes mediates the production of nascent HDL particles. A similar efflux of lipids by ABCA1 and ABCG1 directly in various cells, or indirectly in macrophages (Mφ) of peripheral tissues, contributes to the RCT. LCAT, which catalyzes the esterification of free cholesterol and interaction through CETP transferring cholesterol esters to very low density lipoproteins (VLDL) and low density lipoproteins (LDL) and the phospholipid transfer protein (PLTP) transferring phospholipids from VLDL lipoproteins to HDL, leads to maturation and remodeling of HDL particles. Binding of HDL particles to SR-BI, expressed in hepatocytes, transfers cholesterol esters and other lipids, so that excess cholesterol can be accepted by the liver, catabolized, and excreted via the bile to the intestine. Also, binding of HDL remnants produced after the action of endothelial lipase, or lipid-poor ApoA-I to the beta chain of ATP F1 synthase, expressed at the cell membrane of hepatocytes and other cells (called, also, ecto-F1F0-ATPase that is similar to the F1F0 inner mitochondrial membrane protein complex) promotes cell internalization of HDL particles bound to SR-BI. Abbreviations for various receptors and enzymes are explained in the main text.

**Figure 2 cancers-11-01097-f002:**
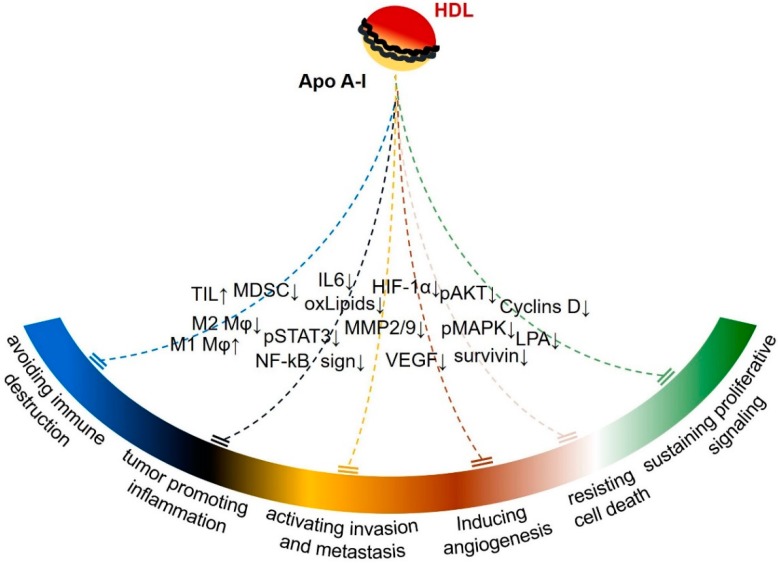
The antitumor activity of ApoA-I in relation to the proposed hallmarks of cancer. Cancer hallmarks affected by ApoA-I, in association with some of the corresponding molecular or cellular mediators reported by various in vitro and animal studies, are shown. It is possible that some mediators may affect more than one hallmark and additional hallmark features, not yet investigated, such as deregulated cellular energetic and genome instability and mutation, may be affected by the tumor suppressive activity of ApoA-I. Abbreviations for various molecules and cell types are explained in the main text. ↑ increase; ↓ decrease.

**Table 1 cancers-11-01097-t001:** Clinicopathologic associations of Apo A-I in cancer.

Organ		Type of Cancer	Association of ApoA-I Levels with:				References
			Risk for the Development of Cancer	Cancer at Primary Diagnosis	Cancer Progression/Metastasis	Cancer Prognosis	
head & neck	squamous cell cancer				+		[79]
	nasopharyngeal carcinoma				−	−	[80,81,82]
lung	non-small cell carcinoma		−	−	− (+)	−	[74,83,84,85,86,87]
esophagus	squamous cell carcinoma			−		−	[88,89]
stomach	gastric cancer			− (+)			[90,91]
colon	adenocarcinoma		−	−	− (+)	−	[76,92,93,94,95,96,97]
liver	hepatocellular carcinoma			−	−	−	[69,70,98,99,100,101,102] [69,70,98,99,100,101,102]
gallbladder	adenocarcinoma			−			[103]
pancreas	adenocarcinoma			−			[104,105]
breast	adenocarcinoma		− (+)	−	−	−	[106,107,108,109,110,111,112,113,114,115,116]
ovary	ovarian carcinoma			−		−	[117,118,119,120,121,122]
uterus	endometrial carcinoma			−			[123,124,125]
cervix	cervical squamous cell carcinoma			−	−		[126,127]
prostate	adenocarcinoma		−	−			[78,128,129]
bladder	transitional cell carcinoma,			− (+)	−	−	[130,131,132,133]
kidney	renal cell carcinoma					−	[134,135]
hematopoietic/lymphoid system	leukemia/lymphoma		−	−	−		[77,136,137]
neural tumors	neuroblastoma			−		−	[138]
	retinoblastoma		+				[139]

− indicates reported inverse association of Apo A-I levels with the specific parameter; + indicates positive association of Apo A-I levels with the specific parameter, reported in a minority of studies or in an isolated study.

**Table 2 cancers-11-01097-t002:** In vitro studies of ApoA-I in cancer.

Type of Cancer	In Vitro System	Apo A-I Manipulation	Biologic Effect and Associated Mechanisms	Ref.
ovarian carcinoma (OC)	murine ovarian cell line ID8	treatment with human ApoA-I or ApoA-I mimetics (L-5F and L-4F)	↓ viability and proliferation	[152]
			↓ LPA-induced viability	
	murine ovarian cell line ID8	treatment with the ApoA-I mimetic D-4F	↓ viability and proliferation	[153]
			↓ oxidative stress	
			↑ MnSOD expression and activity	
	cis-platinum–resistant human ovarian cell lines (OVCAR5, SKOV3, OV2008, and A2780)	treatment with the ApoA-I mimetic L-4F	↓ viability and invasiveness	[154]
			↓ AKT activation	
	cis-platinum-resistant human ovarian cell lines (SKOV3, OV2008)	treatment with the ApoA-I mimetic L-5F	↓ LPA-induced cell viability and VEGF production	[155]
	human ovarian cancer cell lines (OV2008, CAOV-3 and SKOV3)	treatment with the ApoA-I mimetics L-4F and L-5F	↑ proteasome-dependent protein degradation of HIF 1α	[156]
			↓ ROS production	
hepatocellular carcinoma (HCC)	human HCC cell lines (MHCC97H and Huh7)	treatment with recombinant ApoA-I	↓ proliferation (cell cycle arrest)	[100]
			↑ apoptosis	
			↓ MMP2/9	
			↓ VEGF inhibition of the MAPK signaling pathway	
colon adenocarcinoma (CA)	human CA cell lines (DLD-1 and Caco-2) overexpressing ABCA1	transgenic overexpression of ApoA-I, treatment with recombinant ApoA-I or apabetalone (a BET inhibitor, inducer of ApoA-I production)	↓ cell proliferation, migration and invasion	[157]
			modulation of ABCA1 expression through COX-2 downregulation	
			compensation for ABCA1-dependent excessive export of cholesterol	
	murine CA cell line, CT26	treatment with the ApoA-I mimetic L-4F	↓ viability and proliferation	[158]
			↓ cyclin D1 and cyclin A protein levels	
			↓ LPA-induced viability	
breast adenocarcinoma (BA)	human CA cell line, MCF-7	treatment with the ApoA-I mimetic D-4F	↓ oxLDL-induced proliferation	[159]
pancreatic adenocarcinoma (PA)	murine PA cell line P7	treatment with the ApoA-I mimetic L-4F	none	[160]

↑ indicates increase, while ↓ indicates decrease; ABCA1: ATP-binding cassette transporter 1; COX-2: Cyclooxygenase 2; HIF-1α: Hypoxia induced factor 1α; LPA: lysophosphatidic acid; MAPK: Mitogen-activated protein kinases; MMP2/9: Matrix metalloproteinases 2 and 9; MnSOD: Manganese superoxide dismutase; oxLDL: Oxidized low-density lipoprotein; ROS: Reactive oxygen species; VEGF: Vascular endothelial growth factor.

**Table 3 cancers-11-01097-t003:** Animal studies of ApoA-I in cancer.

Type of Cancer	Animal Model	Apo A-I Manipulation	Biologic Effect and Associated Alterations	Ref.
melanoma and non-small lung carcinoma	syngeneic murine melanoma (B16F10L), human melanoma (A375) and Lewis lung (murine) carcinoma cells engrafted subcutaneously or injected intravenously in a metastatic cancer mouse model	human ApoA-I transgenic overexpression or injection of human ApoA-I	↓ tumor growth and metastasis	[161]
			↑ survival	
			↓ tumor angiogenesis	
			↓ MMP-9	
			↓ surviving modulation of the tumor immune microenvironment:	
			↓ M2 Mφ	
			↑ M2 Mφ	
			↓ MDSCs	
			↑ TILs	
		ApoA-I KO	the opposite effects	
ovarian carcinoma	syngeneic murine ovarian carcinoma cells (ID-8) engrafted subcutaneously or injected intraperitoneally in mice	transgenic overexpression of human ApoA-I, or treatment with ApoA-I mimetic peptides (L-5F, L-4F, D-4F)	↓ tumor growth	[152]
			↑ survival	
ovarian carcinoma	syngeneic murine ovarian carcinoma cells (ID-8) engrafted subcutaneously in mice	treatment with ApoA-I mimetic peptides (L-5F, L-4F, D-4F)	↓ tumor growth	[152,153,155,156]
			↓ LPA serum levels	
			↓ tumor angiogenesis	
			↓ VEGF (L-5F)	
			↓ HIF-1α expression (L-4F)	
			↑ MnSOD (D-4F)	
			↓ oxidized phospholipids	
colon adenocarcinoma	AOM/DSS-induced murine colorectal adenocarcinomas	ApoA-I haploinsufficiency Apo A-I^(+/−)^	↑ tumor growth and altered tumor distribution (proximal extension)	[61]
			↓ survival	
			↑ inflammation	
			↑ tumor cell proliferation	
			↑ IL-6, pSTAT3, NF-kB signaling	
colon adenocarcinoma	syngeneic murine colon adenocarcinoma cells CT26 engrafted subcutaneously in mice	treatment with the ApoA-I mimetic peptide L-4F	↓ tumor growth	[158]
			↓ LPA serum levels	
	a murine model for familial adenomatous polyposis (APC^−/+^)		↓ number and size of colon polyps	
colon adenocarcinoma and non-small lung carcinoma	syngeneic murine colon adenocarcinoma (CT26) and Lewis lung carcinoma cells injected intravenously in a metastatic lung mouse carcinoma model	treatment with a concentrate of transgenic tomatoes expressing the ApoA-I mimetic peptide 6F	↓ number of tumors in the lung	[162]
			↓ Notch signaling	
			↓ oxidized phospholipids	
			↑ osteopontin	
			↓ MDSCs in lung and intestine tissues	
colon and ovarian adenocarcinoma	syngeneic murine ovarian carcinoma cells (ID-8) engrafted intraperitoneally and colon adenocarcinoma cells (CT26) injected intravenously in a metastatic lung carcinoma mouse model	treatment with a concentrate of transgenic tomatoes expressing the ApoA-I mimetic peptide 6F	↓ tumor growth in the abdomen	[163]
			↓ number of tumors in the lung	
pancreatic adenocarcinoma	syngeneic murine pancreatic adenocarcinoma cells line P7 orthotopically engrafted in mice	treatment with the ApoA-I mimetic peptide L-4F	↓ tumor growth in the abdomen	[160]
			↓ M2 Mφ in tumors	
breast adenocarcinoma	mammary tumour virus-polyoma middle T-antigen transgenic (PyMT) mice	treatment with the ApoA-I mimetic peptide D-4F	↑ latency of tumor appearance	[159]
			↓ tumor growth	
			↓ oxidized LDL plasma levels	
		transgenic overexpression of human ApoA-I in PyMT mice	none	

↑ indicates increase, while ↓ indicates decrease; AOM: azoxymethane; DSS: dextran sodium sulfate; HIFα: Hypoxia induced factor-1α; LPA: Lysophosphatidic acid; MnSOD: Manganese superoxide dismutase; MMP-9: Matrix metalloproteinases 9; Mφ: macrophages; NF-kB: Nuclear factor kappa-light-chain-enhancer of activated B cells; TIL: tumor infiltrating lymphocytes; MDSC: myeloid-derived suppressor cells; pSTAT3: phosphorylated signal transducer and activator of transcription 3; PyMT: mammary tumour virus-polyoma middle T-antigen transgenic; VEGF: Vascular endothelial growth factor.

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
