# Peer review of "Apolipoprotein A-I (ApoA-I), Immunity, Inflammation and Cancer"

_cancers, 2019, doi:10.3390/cancers11081097_

Round 1
Reviewer 1 Report
Although the Review article of Georgila et al., conceptually sounds at first glance, there are too many shortcuts and inconsistencies that really dampens enthusiasm. A brief overview of major significant issues are listed below but a depth review of all minor issues would be a too long review process.
Major concerns:
The present review is rather a catalog of various inconsistent studies in the field without helping the reader to get ‘a take home message’.
- For instance, in the introduction, the notion of the levels and quality of Apo-I is mentioned but barely discussed in the review (Reference 64 is not even related cancer).
- Paragraph 3 (i.e, anti-inflammatory properties of ApoA-I) seems to be completely disconnected from the other part of the review and does not support the very brief paragraph 7 trying to link anti-inflammatory properties of ApoA-I to its tumor suppressive function. How the anti-inflammatory properties of ApoA-I promotes conversion of TAM from M2 to M1? The use of Abca1 or Abcg1 knockout animals also leads to the same phenotype, which is counterintuitive and not discussed by the authors. Along this line, the authors did not mention that the deficiency of these transporters does not always mirror the ApoA-I effect. This is an issue since the authors use several references with cholesterol transporter deficiency to argue on the beneficial effect of ApoA-1.
- Out of the 182 references, there are only 29 references more recent than 2017 and few from 2016.
- In contrast to the development of ApoA-I mimetic peptides, the failure of recent clinical trials in CVD due to lack of efficacy has not been discussed. How the authors envision a repurposing of these strategies?
- Paragraph 2 is again a catalog of various different notions including at the same time the transcriptional regulation of ApoA-1, its function in RCT or anti-inflammatory functions. It is difficult to read. Unless the authors could link these different notions, it would have been better to separate these notions. How is it link to cancer?
- Paragraph 3. Lack of recent references in the field (role of ApoA-1 in DCs from Tall’s or Norata’s group). Lack of big overview of HDL proteomic (i.e, princep manuscript from Heinecke’s group). How is it linked to free ApoA-1? Are HDL function really altered in cancer?
- Paragraph 4. Although the authors tried to summarize their findings in Table 1, the reader is quickly lost with the successive catalog of association between HDL levels and cancer. Is there any specificity? Why references 126 to 138 are not discussed in the main text? The discussion on response to therapy could have also been dissociated from this paragraph?
- The last paragraph (p6 l254) is quite confusing with again a catalog of negative associations without any rational or explanation.
Paragraph 5. Based on the description of the role of HDL in various cancer or stroma cell lines, it seems that HDL is pleiotropic (Apoptosis, ECM, angiogenesis, migration, viability, arrest,…). However, only 8 references are provided in this paragraph (very limited compared to other ones). What is the ‘take home message’?
- Paragraph 6. The other now come back to preclinical studies, which makes the review difficult to follow. The paragraph of cholesterol transporters (p10 l341) is disconnected from the rest of the paragraph.
Overall, this review may require a general reorganization along with more recent references and should avoid to list oldest references as
Author Response
We thank the reviewer for constructive comments which provided us with the opportunity to improve our manuscript.
-The present review is rather a catalog of various inconsistent studies in the field without helping the reader to get ‘a take home message’.
A take home message is stated in the Conclusion, section p16, lines 496-500, and in figure 2.
- For instance, in the introduction, the notion of the levels and quality of Apo-I is mentioned but barely discussed in the review (Reference 64 is not even related cancer).
In line with the reviewer’s comment, we have added a paragraph in Section 3 (lines 189-199) of the revised manuscript that addresses deregulation of ApoA-I levels in relation to cancer. The paragraph includes appropriate citations. An additional cancer-related reference (64) next to original ref 64 (current 63) have been added.
- Paragraph 3 (i.e, anti-inflammatory properties of ApoA-I) seems to be completely disconnected from the other part of the review and does not support the very brief paragraph 7 trying to link anti-inflammatory properties of ApoA-I to its tumor suppressive function. How the anti-inflammatory properties of ApoA-I promotes conversion of TAM from M2 to M1? The use of Abca1 or Abcg1 knockout animals also leads to the same phenotype, which is counterintuitive and not discussed by the authors. Along this line, the authors did not mention that the deficiency of these transporters does not always mirror the ApoA-I effect. This is an issue since the authors use several references with cholesterol transporter deficiency to argue on the beneficial effect of ApoA-1.
We thank the reviewer for his/hercomment.Wewould like to point to the fact thattherole of ApoA-I in immunity and inflammationrepresentsone of thegoalsof this review, as also highlighted in the title.Innew Section 2 of the revised manuscript, we have added a sentence to indicatethatthe anti-inflammatory properties of ApoA-Imay impinge on cancer-related processes (lines 183-185), thus further linking this paragraph with the rest of the review.
We agree with the reviewer on the counterintuitive nature of abc/g1 and ApoA-I tumor suppressive and anti-inflammatory properties. As suggested, we have added a comment that ‘the deficiency of these transporters does not always mirror the ApoA-I effect’ and discuss further this point in the revised section 8 (lines 469-486). The precise mechanism by which ApoA-I promotes conversion of TAM from M2 to M1 has not been defined, as stated in lines 397-8.
- Out of the 182 references, there are only 29 references more recent than 2017 and few from 2016.
We note that 36,8% of the references in the revised manuscript are from the last 5 years. As suggested result of the revision, we have introduced 20 additional recent references (numbers 16, 29, 56, 58, 64, 65, 67, 71, 85, 88, 103, 133, 137, 138, 139, 144, 166,194, 198, 199;References with bold letters represent studies of 2019).
- In contrast to the development of ApoA-I mimetic peptides, the failure of recent clinical trials in CVD due to lack of efficacy has not been discussed. How the authors envision a repurposing of these strategies?
As suggested, we discuss and comment on this issue in new section 7 (lines ‘’439-445’’)
- Paragraph 2 is again a catalog of various different notions including at the same time the transcriptional regulation of ApoA-1, its function in RCT or anti-inflammatory functions. It is difficult to read. Unless the authors could link these different notions, it would have been better to separate these notions. How is it link to cancer?
Paragraph 2 of the original submission intended to provide a brief introduction into the expression and physiological roles of ApoA-I. To emphasize this fact in the revised manuscript, we have merged paragraphs 1 and 2 under the same title (“Introduction”) and have re-organized the information provided herein. We have also added a paragraph in revised Section 3 (lines 191-199) to link this introductory information to cancer. Finally, we have re-organized Section 6 to include known and putative contributions of ApoA-I/RCT to cancer.
- Paragraph 3. Lack of recent references in the field (role of ApoA-1 in DCs from Tall’s or Norata’s group). Lack of big overview of HDL proteomic (i.e, princep manuscript from Heinecke’s group). How is it linked to free ApoA-1? Are HDL function really altered in cancer?
Tall’s studies about the connection of ApoA-I with inflammation were included in the original manuscript (original ref 49, 58; current 48 and 57 respectively). As suggested, additional references from Tall’s and Norata’s groups are added in the revised manuscript (ref 29, 56, 58, 65). HDL proteomics analysis is outside the scope of this review. However, a recent review from Heinecke’s group is also added in the references (reference 16), as suggested.
- Paragraph 4. Although the authors tried to summarize their findings in Table 1, the reader is quickly lost with the successive catalog of association between HDL levels and cancer. Is there any specificity? Why references 126 to 138 are not discussed in the main text? The discussion on response to therapy could have also been dissociated from this paragraph?
In order the reader to follow easier the text we advanced the introduction of table 1 (in the beginning of the revised section, lone 201)References 126 to 138 (current 107 and 108) are included in the revised main text (lines 240).
We believe that the response to therapy is closely connected to the prognostic information associated with ApoA-I and, therefore is described in the same paragraph of the revised text (paragraph 3,pp.6lines235-252)and further commented in the last paragraph of the section (lines 274-279).
- The last paragraph (p6 l254) is quite confusing with again a catalog of negative associations without any rational or explanation.
As suggested, we introduced a comment about the limited number of reported positive associations between ApoA-I and cancer inpp..6lines270-273(. ‘’ It is unclear whether the positive correlation between ApoA-I levels and cancer parameters reported in a minority of studies are specific reflecting, in these particular situations, tumor promoting processes, associated with increased cholesterol uptake of malignant cells through the HDL/SR-BI pathway, as proposed by some studies ’’).
-Paragraph 5. Based on the description of the role of HDL in various cancer or stroma cell lines, it seems that HDL is pleiotropic (Apoptosis, ECM, angiogenesis, migration, viability, arrest,…). However, only 8 references are provided in this paragraph (very limited compared to other ones). What is the ‘take home message’?
The number of in vitro mechanistic studies about ApoA-I and cancer phenotype is relatively limited in the literature, as the reviewer correctly points out. The ‘’take home massage’’ of the paragraph is stated at the beginning of the revised section (new Section 4, lines 284-286) ‘’.
- Paragraph 6. The other now come back to preclinical studies, which makes the review difficult to follow. The paragraph of cholesterol transporters (p10 l341) is disconnected from the rest of the paragraph.
We believe the separation of the in vitro and in vivo studies emphasize the findings summarized in figure 2 about cell-autonomous and non-autonomous mechanisms involved in the tumor suppressive activity of ApoA-I,as discussed in the text 447-449,Section 8). As suggested, we transferred this particular paragraph about the cholesterol transporters in the revised section 8, lines 471-474 in order not to appear disconnected
Reviewer 2 Report
This is an interesting study showing the potential antitumorigenic effects of apoA-I and its association with apoAI-mediated effects on immunity and inflammation. The review article is timely, informative, comprehensive and well organized. To improve the quality of work, I have enclosed a few points that should be included.
Table 1
+ indicates reported inverse association of Apo A-I levels with the specific parameter; # indicates positive association of Apo A-I levels with the specific parameter.
This is rather confusing, I would include an inverse association with a negative symbol (–) and positive (+) or indicate “inverse association” and positive association”.
The bases and background of apoA-I mimetics used in this pathology should be included in the Introduction (rather than include it in the point 8).
Revilla et al have recently reviewed the role of ABCA1 in ETM processes in Int J Mol Sci. 2019 May 18;20(10). This point should be included.
A recent review of Cedó et al has been published in J Clin Med. 2019 Jun 14;8(6).on the role of HDL in breast cancer. The authors have expanded these studies to other cancers and focused on apoA-I. Clarify this point.
The authors should clarify that HDL-associated cholesterol could promote some tumorigenic processes (beyond apoA-I actions). Cancer cells may take up cholesteryl ester thought HDL/SR-BI pathway and enhance malignant phenotypes Gutierrez-Pajares et al. Front. Pharmacol., 2016 |doi.org/10.3389/fphar.2016.00338.
Author Response
We thank the reviewer for constructive comments which provided us with the opportunity to improve our manuscript
Table 1
- + indicates reported inverse association of Apo A-I levels with the specific parameter; # indicates positive association of Apo A-I levels with the specific parameter.
This is rather confusing, I would include an inverse association with a negative symbol (–) and positive (+) or indicate “inverse association” and positive association”.
As suggested, in our revised manuscript a negative (–) and positive (+) symbol are used for indicating inverse and positive association, respectively.
-The bases and background of apoA-I mimetics used in this pathology should be included in the Introduction (rather than include it in the point 8).
As suggested, we introduced the base about ApoA-I mimetics in the revised introduction (p.2 lines 60-62)leaving further analysis for the revised section 7.
-Revilla et al have recently reviewed the role of ABCA1 in ETM processes in Int J Mol Sci. 2019 May 18;20(10). This point should be included.
As suggested, we included this recent reference (nu 198) and commented on it in the revised section 8, lines 470-474 (‘For example, it has been shown that mice deficient in ABCG1 and ABCA1, when fed a “western”-type diet, display reduced growth of tumors derived from subcutaneously engrafted melanoma or bladder carcinoma cells, while other studies have attempted to associate ABCA1 with epithelial mesenchymal transition in breast cancer’’ ).
-A recent review of Cedó et al has been published in J Clin Med. 2019 Jun 14;8(6). on the role of HDL in breast cancer. The authors have expanded these studies to other cancers and focused on apoA-I. Clarify this point.
As suggested, we included this reference (Ref number 166) and commented on it in lines 347-348 (‘Interestingly, dysfunctional, oxidized ApoA-I/HDL has been reported to promote breast cancer metastasis in mice [166].’’)
-The authors should clarify that HDL-associated cholesterol could promote some tumorigenic processes (beyond apoA-I actions). Cancer cells may take up cholesteryl ester thought HDL/SR-BI pathway and enhance malignant phenotypes Gutierrez-Pajares et al. Front. Pharmacol., 2016 |doi.org/10.3389/fphar.2016.00338.
As suggested, we included this reference (number 138) and commented on the possible tumor promoting role of HDL/SR-BI in the revised section 3 (lines 270-273 ‘’It is unclear whether the positive correlation between ApoA-I levels and cancer parameters reported in a minority of studies are specific reflecting, in these particular situations, tumor promoting processes, associated with increased cholesterol uptake of malignant cells through the HDL/SR-BI pathway, as proposed by some studies [137-139] ’’)
Reviewer 3 Report
This review by Drakos et al sheds light on many important functions of Apo-A1 as they are related to inflammation and cancer. Much research in the past has been devoted to antiatherogenic properties but this work does a good job of putting forward other important functions of the protein (by citing appropriate references) which may result in the development of therapeutic approaches for several inflammatory/autoimmune diseases and cancers. Clearly, much work needs to be done in this area to investigate the therapeutic role of Apo-A1 in immuno-oncology and inflammation.
I noticed a few spelling errors.
I recommend publishing this work
Author Response
We thank the reviewer for constructive comments which provided us with the opportunity to improv
-I noticed a few spelling errors.
As suggested, after careful proofreading, we have corrected spelling errors.
e our manuscript
Round 2
Reviewer 1 Report
The authors have answered most of my comments. The review could have benefit from a discussion on CETP inhibitors and the failure of recent clinical trials in CVD. However, the authors dampened their conclusions of the role of ApoAI in cancer